# Comparison of the Total Polyphenol Content and Antioxidant Activity of Chocolate Obtained from Roasted and Unroasted Cocoa Beans from Different Regions of the World

**DOI:** 10.3390/antiox8080283

**Published:** 2019-08-06

**Authors:** Bogumiła Urbańska, Jolanta Kowalska

**Affiliations:** Faculty of Food Sciences, Department of Biotechnology, Microbiology and Food Evaluation, Warsaw University of Life Sciences, 159 Nowoursynowska str., 02-787 Warsaw, Poland

**Keywords:** cocoa, chocolate, polyphenols, antioxidants

## Abstract

The polyphenol content of cocoa beans and the products derived from them, depend on the regions in which they are grown and the processes to which they are subjected, especially temperature. The aim of the study was to compare the total content of polyphenols and antioxidant activity of chocolates obtained from roasted and unroasted cocoa beans. The chocolates produced from each of the six types of unroasted beans and each of the five types of roasted beans were investigated. The seeds came from Ghana, Venezuela, the Dominican Republic, Colombia and Ecuador. The highest total polyphenol content was determined in cocoa beans originating from Colombia and in the chocolates obtained from them. A higher content of total polyphenols was found in unroasted cocoa beans, which indicates the influence this process had on the studied size. The ability to scavenge free DPPH radicals was at a high level in both the beans and the chocolates produced from them, irrespective of the region where the raw material was grown. A positive correlation between the total polyphenol content and the ability to scavenge free radicals was found.

## 1. Introduction

Chocolate is one of the most valued food products in the world [1]. Bitter chocolate in liquid form was already discovered in South America over 3000 years ago. This valuable product is made from cocoa liquor and fat (cocoa butter), with the addition of sugar, as well as milk and other additives (depending on the type of chocolate).

The main ingredient of chocolate is cocoa liquor, which is a mixture of fat and non-fat ingredients from the processing of cocoa beans. Furthermore, chocolate is produced with the addition of cocoa powder, which was recognised by EFSA as one of the richest sources of polyphenols [2]. Therefore, cocoa has recently become the target of increased scientific research, due to its pro-health properties.

Fresh cocoa beans contain about 32–39% water, 30–32% fat, 10–15% protein, 5–6% polyphenols, 4–6% pentosans, 2–3% cellulose, 2–3% sucrose, 1–2% theobromine, 1% acids and less than 1% caffeine. It is also a rich source of mineral components [3,4].

Three types of flavonoids dominate in cocoa beans: Proanthocyanins (circa 58%), catechins or flavan-3-ols (circa 37%) and anthocyanins (circa 4%) [5,6].

The high polyphenol content of cocoa, combined with its wide presence in many food products, makes it particularly interesting from a nutritional and health point of view [6,7,8,9,10,11]. 

The content and composition of polyphenols differs depending on the genotype, origin, growth conditions, degree of ripeness of the cocoa fruit and the grain processing parameters [12,13,14,15]. Knowledge of the changes in the polyphenol content throughout the technological chain, enables us to take steps to maintain the highest possible polyphenolic content in the final product.

Polyphenols shape not only antioxidant properties, but also affect sensory properties such as colour and taste [14,16,17]. During all stages of processing, the polyphenols present in cocoa beans may undergo many transformations, including polymerization, hydrolysis or reactions with proteins [18].

Roasting is a technological process which, due to its high temperature, is one of the most important in shaping the quality and sensory properties of cocoa beans and the products derived from them. During roasting, the structure of the beans is dried and loosened to remove the husks. Grinding of the kernels and pumping of the cocoa fat follows [19].

The roasting time lasts from 5 to 120 min (usually 10 to 35 min), at temperatures ranging from 110 °C to 160 °C (usually 120 °C−140 °C) [7,14,20]. High temperatures and the dehydration of cocoa beans in the roasting process, reduce the concentration of polyphenols and many volatile acids, especially acetic acid, which is responsible for the acidity of the product, as well as the bitter and astringent taste [21,22].

The roasting process also stimulates protein degradation, the synthesis of sulfur compounds, the Maillard reaction and the caramelization of sugars. These reactions allow new compounds to form, which contributes to the characteristic aroma, taste and colour of chocolate [23,24]. Oracz, et al. [25] stated that during the roasting process, conditions such as time and temperature influence phenolic stability, as well as the characteristics of the obtained taste.

Cocoa beans that are only fermented and dried (not roasted, referred to as "raw") during the production process contain many more phenolic compounds, which can have a positive effect on human health [18]. Omitting the roasting process, and not using high temperatures during the production of raw chocolates, results in the inability to evaporate many volatile components from the grain, including acetic acid, as well as increasing the acidity of such grains. The production of this type of chocolate does not use alkalization either, so the process of conching is often extended to up to four days, which makes it possible to obtain a final product that has a structure similar to traditional chocolates [26,27].

Roasting whole beans requires more energy than roasting smaller pieces, and at the same time makes it difficult to roast the beans evenly. Differences in roasting greatly influence the final product; i.e., the properties of chocolate, as well as its sensory and physicochemical properties. That is why manufacturers often roast a shot; i.e., ground beans. In this study, the antioxidant properties of beans that were roasted whole and ground, originating from the same region, were compared.

A number of studies are currently underway to develop a technology for roasting cocoa beans and cocoa middlings that will result in a semi-finished product with the least possible reduction in bioactive compounds and the lowest possible content of anti-nutritional compounds, such as acrylamide and acrolein [28].

The aim of this study was to compare the total polyphenol content and its ability to inactivate free DPPH radicals in chocolates produced from roasted and unroasted cocoa beans from different regions.

## 2. Methodology

### 2.1. Experimental Material

The experimental material consisted of roasted and unroasted cocoa beans, as well as the bitter chocolate obtained from the beans, from the Dominican Republic, Ghana, Venezuela, Colombia and Ecuador. Presented in Table 1 is a list of the research material and its labelling. The beans came from the 2014 harvest and were submitted for testing by one of the chocolate producers in Poland.

In order to illustrate the appearance of roasted and unroasted beans from different regions of the world, photographs were taken (shown in Figure 1).

### 2.2. Analytical Methods

Chemical analyses have been carried out in at least three parallel repetitions. Total polyphenol content was determined by Folin-Ciocialteu’s method [28]. Based on preliminary tests, a 70% acetone solution was used as a solvent to prepare extracts.

The extracts were prepared by weighing about 5 g of crushed test material into 300 mL grinding conical flasks and adding 100 mL of 70% acetone (*v*/*v)*.

The samples were then shaken for 30 min in a Multi-Shaker PSU 20 Biosan shaker. Following this procedure, the solutions were filtered through the corrugated filters into 100 mL grinding flasks. In order to determine the total polyphenol content, 300 µL of the extract was taken from the tubes, and 4.15 mL of deionized water, 500 µL of 20% sodium carbonate solution and 50 µL of Folina-Ciocialteu reagent were added.

The blank sample was prepared by sampling: 300 µL extraction solution, 4.15 mL deionized water, 500 µL 20% sodium carbonate solution and 50 µL Folina-Ciocialteu reagent. Absorbance was measured at 700 nm on a SHIMADZ UV-1201V spectrophotometer. The apparatus was zeroed to a blank.

In order to calculate the total polyphenol content, a standard curve was prepared. Based on the results obtained, the graphical dependence of the absorbance of the solution on the amount of gallic acid contained in it was plotted. The total polyphenol content was calculated on the basis of the calibration curve and expressed in gallic acid equivalent per 100 g of product. 

Determinating the ability of extracts to inactivate stable DPPH radicals 

The extracts were prepared by weighing them into 300 mL grinding conical flasks of about 5 g of crushed test material and adding 100 mL of 70% acetone (*v*/*v*). The samples were then shaken for 30 min in a Multi-Shaker PSU 20 Biosan shaker. Following that, the solutions were filtered through the corrugated filters into 100 mL grinding flasks. Acetone extract (4 mL) and DPPH solution (1 mL), were taken to determine the appropriate sample. Acetone extract (4 mL) and methanol (1 mL), were collected for the blank sample. The samples were mixed and left to stand for 30 min, then absorbance was measured on the NOVASPEC II Pharmacia spectrophotometer (zeroing the apparatus for the blank test) at a wavelength of 562 nm in glass cuvettes with a diameter of 1 cm [29].

The antioxidant activity of the extracts against DPPH was calculated using the formula:Act. = [(Ak − Awł)/Ak] × 100%.(1)
where Act.-ntioxidant activity (%); Ak—the absorbance of the control sample; and Awł—absorption of the specific sample.

Microsoft Excel 2013 for Windows 10 was used to calculate the average values of the obtained results and to create graphs. Statistical analysis of the obtained results, correlation tests and the significance of the differences between the test samples were carried out using Statistica 13.0 using Tukey’s test at the level of significance *p <* 0.05.

## 3. Results and Discussion

### 3.1. Determination of Total Polyphenols in Cocoa Beans

A total content of polyphenols in the research material was determined. The results were calculated in terms of gallic acid equivalent, as shown in Figure 2. The results for roasted cocoa beans in their entirety (marked with the letter “r”), middlings-crushed cocoa beans (marked with the letter “c”) and whole unroasted cocoa beans (marked with the letters “ur”) were presented.

The highest polyphenol content was found in both roasted and unroasted cocoa beans originating from Colombia, respectively 3781 mg/100 g of product and 3766 mg/100 g of product. The lowest total polyphenol content was found in roasted and unroasted cocoa beans originating from Venezuela (996 and 1034 mg/100 g of product). These differences can be explained by factors such as plant variety, geographical region, degree of maturity and post-harvest conditions [24]. The study also included cocoa beans that were organically farmed from Peru; ones which are used only in unroasted forms to produce raw chocolates. The polyphenol content of these beans was determined to be 2778 mg, the second highest after that of Colombia. The polyphenol content of roasted cocoa meal was also analysed, and the results obtained for whole roasted beans were compared. In the majority of analysed samples, the polyphenol content in crushed beans was significantly lower. The differences could result from a greater loss of these compounds due to the influence of oxygen and light on a smaller surface area. Additionally, the interior of the grain was discovered, to some extent, as a result of grinding, which additionally facilitated the influence of external factors on phenolic compounds (Figure 2).

In a study by Salvador, et al. [30], cocoa beans were analysed during production at one of the production plants in Brazil. In raw beans, over 6000 mg/100 g of product was determined, while in roasted beans from the same production cycle only about 1050 mg/ 100 g of product was determined. 

In all types of unroasted grains, except for Colombia’s (a similar level was determined), a higher content of polyphenols in grain was found, compared to the amount in roasted grains. The biggest difference was discovered in the beans from Ghana, and was almost 30% higher before roasting, whereas in Venezuela the difference was about 4% in favour of unroasted beans.

There are many known cases in literature where these differences are much greater. According to Medeiros, et al. [8], as a result of technological parameters and operations in the final product, the loss of flavonoids derived from cocoa beans can reach up to 80% of their initial content.

According to Gültekin-Özgüven, et al. [31] roasting and alkalization of cocoa beans reduces polyphenolic content by 65% and 87% respectively.

The determined polyphenol content may also be affected by the type of extraction used, the length of the procedure, the solvent used and the degree of fragmentation of the research material [32,33].

Statistical analysis showed significant differences in the content of polyphenols with a confidence level of 95%. Additionally, through the analysis of variance (ANOVA) for the one-way experiment, it was found that the type of cocoa beans (*p*-Value < 0.05) had a significant effect on the content of polyphenols. The beans from Colombia and Peru, which are separate homogeneous groups and have the highest polyphenolic compound content, deserve special attention. Based on the Tukey HDS test, eight homogeneous groups, marked with the same letters in Figure 2, were distinguished from cocoa beans. 

The regression analysis showed a positive correlation (0.36) between the type of cocoa beans used in production and the content of polyphenols in chocolate.

### 3.2. Determination of Total Polyphenols in Chocolates

The results of total polyphenolic content in chocolates converted to gallic acid equivalent are presented in Figure 3. The polyphenolic content in chocolates ranged from 910 mg/100 g of chocolate product produced from roasted beans from Venezuela to 4055 mg/100 g of chocolate product from roasted beans from Columbia. The most chocolates had a higher polyphenol content than the amounts indicated in the cocoa beans. The process of chocolate production is complex and depends on many factors. The main source of phenolic compounds is the raw material-cocoa beans. According to literature data, most polyphenols undergo degradation during high-temperature processes. It should be remembered, however, that some polyphenols, e.g, (−)-epicatechin, may form complex, insoluble complexes, which are very difficult to determine analytically. At the same time, due to high temperatures, complex procyanidins may degrade to monomers, which in turn are determined analytically, thus affecting the overall polyphenol content of the product. In addition, bitter chocolates are often accompanied by the addition of a skimmed cocoa powder, which is an excellent source of polyphenols, and their positive health effects have been confirmed in the EFSA report [2].

Cocoa crops in Colombia currently have a high genetic variability, due to the crosses that can be found in some cocoa varieties (Stranger/Amazonian and Trinitario clones) [34]. As a result, the ecoclimatic conditions in the territory of Colombia are conducive to crop expansion into the country, and the increased capability to produce cocoa ecotypes with different bioactive profiles and flavours that are classified as high quality beans [34].

Polyphenols are contained in the non-fat components of cocoa beans, so it should be remembered that increasing the proportion of cocoa liquor results in an increase in the content of polyphenols in chocolates [35]. Jabłońska-Ryś, et al. [35] determined 2241–2746 mg of polyphenols per 100 g of product in dessert chocolate containing 70% cocoa, while in bitter choclates containing 75–80% cocoa, 2164–3129 mg of polyphenols were found per 100g of product.

In the present study, only chocolate from Venezuelan beans (910 mg/100 g product) showed lower results than those reported in the literature. In other cases, values higher than those presented in the studies by Jabłońska-Ryś, et al [35] or Kowalska and Sidorczuk [36] were obtained. The studies conducted by Meng, et al. [37] confirmed the influence of cocoa liquor’s mass in chocolate on the content of polyphenols (578 mg/100 g in dark chocolates, while in milk and white chocolates the amount was 160 and 126 mg/100 g respectively). Studies by Cooper, et al. [38] also showed that apart from one chocolate from Venezuelan beans, the content of polyphenols was lower than in this study. On the other hand, Żyżelewicz, et al. [18] showed that dark chocolates produced with and without the addition of cocoa mass, prepared from unroasted cocoa beans, contained a higher concentration of total polyphenols (360 mg/100 g) compared to chocolate produced on the basis of roasted beans, in which only 841 mg of polyphenols per 100 g of the product was determined. The authors claim that during the preparation of chocolate, polyphenols may undergo many transformations, including polymerization and hydrolysis, as well as interacting with proteins and the products of the Maillard reaction.

A study by Lucia Godočiková, et al. [39] noted that chocolate produced in the traditional way (the roasting stage included) had almost twice the polyphenol content compared to cold processed products. Polyphenols can break down (degrade) or condense into complex compounds as a result of high temperatures. The lack of a roasting stage, for which the temperature of 110–160 °C is applied, may reduce the content of polyphenolic compounds, which was confirmed by Todorovic, et al. [40] in their studies. The tests were carried out for chocolates containing 65 to 75% cocoa. In the Jalil and Ismail [41] studies, chocolates obtained from unroasted beans were characterized by indirect polyphenol content in comparison to other analysed products. The amount of the analysed component in chocolates confirmed that the content of bioactive compounds depends on the technological process, raw material composition of the final product and the origin of cocoa beans [41].

The amount of polyphenols determined could also be affected by the type of extraction used, extraction time, the solvent used and the degree of fragmentation of the research material. Benayad, et al. [42], Cheng et al. [43] and Boulekbache-Makhlouf, et al. [44] showed that the use of acetone, in comparison to other polar organic compounds, increased the extraction of flavonoids and flavonoids from different plant materials.

The results of many scientific studies show how complex the process of determining the polyphenols content in chocolates is. Therefore, it is not clear which cocoa beans are the best source of polyphenols.

The statistical analysis of chocolates in the range of polyphenol content showed that only in one case (products from Venezuela), there was no statistically significant difference between the size determined in chocolates and beans (Figure 3). The remaining chocolates differed significantly from each other statistically with a confidence level of 95%. The analysis of variance (ANOVA) for the one-way experiment showed that the region of origin of cocoa beans significantly influenced the content of polyphenolic compounds in chocolates obtained from them (*p*-value < 0.05). Based on the Tukey HDS test, eight homogeneous groups, marked with the same letters on the data labels, were distinguished among chocolates.

Based on the results obtained in the present study and literature data, it can be concluded that the content of polyphenols depends on many factors, both those resulting from the genotype and region of raw material cultivation, as well as the technological processes and parameters used. Despite cultivation of the same varieties under similar conditions and the application of similar processing conditions, the content of polyphenols varies. It should be remembered that plants produce polyphenols in response to stresses, which can be very strong sunlight, drought, pest infestation and many others. In addition, the polyphenol content varies depending on the period of harvest. Cocoa beans from different plantations are mixed, and further processed as such. Therefore, there are differences in the polyphenols content of both the raw materials and the products derived from them. The analysis of the polyphenol content in the beans from different regions, taking into account the applied technological processes and their parameters, may be useful in creating blends and optimising the quality characteristics of the chocolates obtained from them.

### 3.3. Determination of the Ability of Extracts to Inactivate Stable DPPH Radicals in Cocoa Beans and Chocolates Derived Therefrom

The main polyphenols in cocoa beans are catechins, epicatechins, anthocyanins and procyanidins, the presence of which affets antioxidant activity [28].

Figure 4 and Figure 5 show the results of the analysis of the activity of the tested extracts against DPPH radicals. The results of antiradical activity of antioxidant compounds in relation to DPPH radicals were calculated on the basis of measured absorbance values.

Both the beans (roasted and unroasted) and chocolates produced from them were characterized by high ability to scavenge stable DPPH radicals. The analysis showed the influence of the roasting process on the antioxidant activity. In all roasted cocoa beans the ability to extinguish stable DPPH radicals was lower. Moreover, contrary to the determination of polyphenol content, higher antioxidant activity was determined in roasted meal in most of the analyzed samples than in whole grains. Referring to the results obtained for raw chocolate from Peru beans, it should be noted that despite the high content of polyphenols, the ability to extinguish the stable DPPH radicals was indirect. This is further evidence that the antioxidant properties of chocolate are an extremely complex phenomenon, dependent on a number of factors, and therefore it is not possible to determine the polyphenolic content and antioxidant activity unequivocally. Despite the separation of homogeneous groups, no statistically significant differences in antioxidant activity against stable DPPH radicals were found for cocoa beans studied, which was confirmed by statistical analysis (*p*-value = 0.148). The lowest antioxidant activity to extinguish DPPH radicals was obtained for chocolates from Ecuador (90.75%), Colombia (91.91%) and Dominican Republic (90.93%). Particularly noteworthy is chocolate from Colombia, which was characterized by the highest content of polyphenols in total. These compounds, according to literature data and the results of the correlation carried out in this study, significantly shape the antioxidant activity.

The above data show that traditional roasting significantly reduces both the concentration of polyphenols and the antioxidant activity of cocoa beans. These observations are consistent with the reports of Djikeng, et al. [45] who demonstrated that the decrease in antioxidant activity of cocoa beans during roasting was associated with the destruction of polyphenols contained in them. Roasting is considered one of the stages in the processing of cocoa beans that leads to a high loss of phenolic compounds and a decrease in antioxidant activity, as confirmed by Bauerin, et al. [46]. Similar conclusions were reached by Arlorio, et al. [47], who compared the antioxidant capacity of roasted and unroasted cocoa beans. Hu, et al [48] reported a decrease in antioxidant activity of between 44 and 50% from high-temperature roasting (190 °C). According to Gültekin-Özgüven, et al. [32], roasting cocoa beans reduced the antioxidant capacity of DPPH by 21% and of ORAC by 51%. According to the results of the Kowalska and Sidorczuk [38] studies, the tested cocoa beans and chocolates were characterized by a high ability to scavenge free radical DPPH at the level of 88–92%. These results are slightly lower than the results obtained in this study (cocoa beans above 95%, and chocolate from 90% to 96%), which may be due to the different content of cocoa components and the region of origin of the beans. Żyżelewicz et al. [49], in their studies, showed that the antioxidant activity of chocolate decreased with an increase in the percentage of prepared cocoa liquor from unroasted beans. Wollgast, et al. [50] found that the evaluation of polyphenolic compounds and antioxidant activity depends largely on the solvent and extraction procedure, which is not standardized in the cocoa literature, so the data are difficult to compare. According to Di Mattia, et al. [29], discrepancies in total phenol (TPC) colorimetric tests may occur due to phenolic compounds being used as reference for the standard curve and the presence of reducing compounds that interfere with the test. Therefore, comparison of antioxidant activity results may be problematic due to the large number of heterogeneous tests used.

The regression analysis at a 95% confidence interval showed that there is a positive correlation (0,86) between the content of polyphenols in chocolate and the ability to inactivate stable DPPH radicals. Additionally, a very strong correlation (r = 0.72) between the antiradical activity of grain extracts from different regions of the world and chocolate extracts produced from them was demonstrated on the basis of a statistical analysis.

## 4. Summary and Conclusions

Based on the research conducted and results obtained, it is now known that roasted cocoa beans in most of the analyzed samples had a lower polyphenol content than unroasted grains. However, the content of polyphenols in chocolates was much higher than in the cocoa beans from which they were obtained. The ability to extinguish free DPPH radicals was at a high level both in the beans and chocolates, and was higher than in the literature’s data. This ability decreased after roasting the cocoa beans, and after the whole process of chocolate production, in relation to the beans from which it was produced. 

The conducted research showcases the influence of the type of cocoa beans and the technological processes used on the properties of the chocolates obtained. Further analysis of chocolate products based on unroasted beans and the evaluation of their usefulness, in terms of dietary inclusion of products with specific antioxidant properties, seems justified.

## Figures and Tables

**Figure 1 antioxidants-08-00283-f001:**
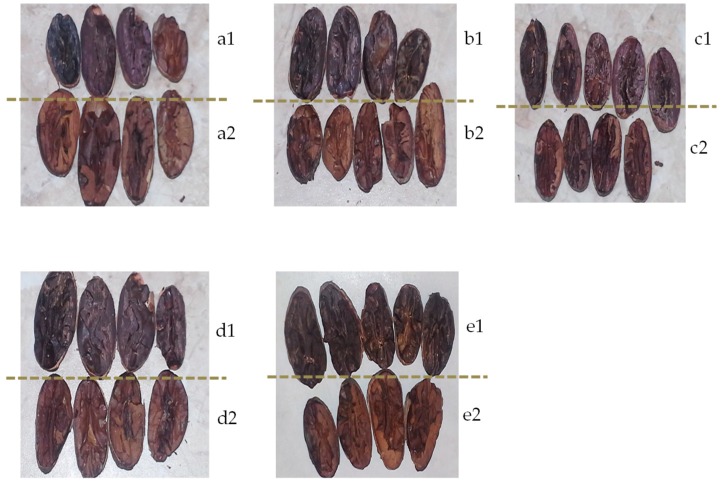
Cocoa bean cross-section photos (taken by authors); (**a1**) unroasted Colombia, (**a2**) roasted Colombia, (**b1**) unroasted Dominican Republic, (**b2**) roasted Dominican Republic, (**c1**) unroasted Ecuador, (**c2**) roasted Ecuador, (**d1**) unroasted Ghana, (**d2**) roasted Ghana, (**e1**) unroasted Venezuela, (**e2**) roasted Venezuela.

**Figure 2 antioxidants-08-00283-f002:**
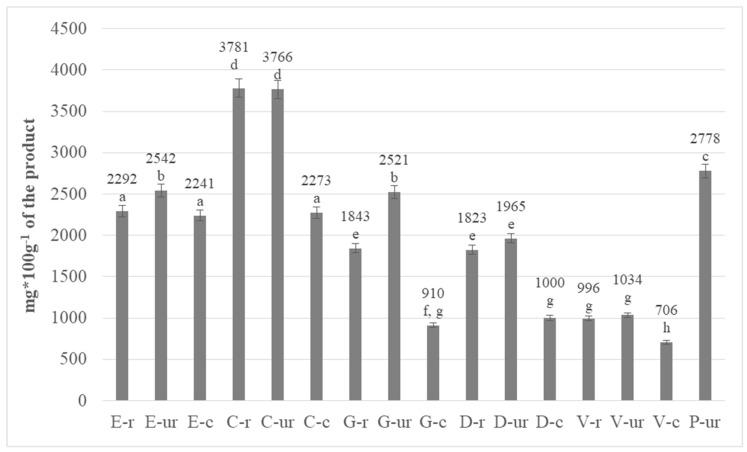
Total polyphenolic content in roasted and unroasted cocoa beans from different regions of the world (the same letter means no statistically significant differences between the analysed products at the level of significance α = 0.05; abbreviations used in the graph are described in Table 1).

**Figure 3 antioxidants-08-00283-f003:**
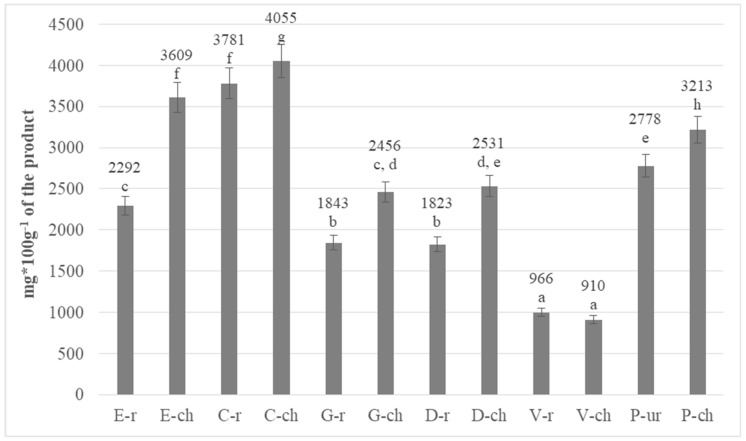
Comparison of the total polyphenol content in beans from different regions of the world and in the chocolates obtained from them (the same letter means that there are no statistically significant differences between the analysed products at a confidence level of α = 0.05; the abbreviations used in the graph are described in Table 1).

**Figure 4 antioxidants-08-00283-f004:**
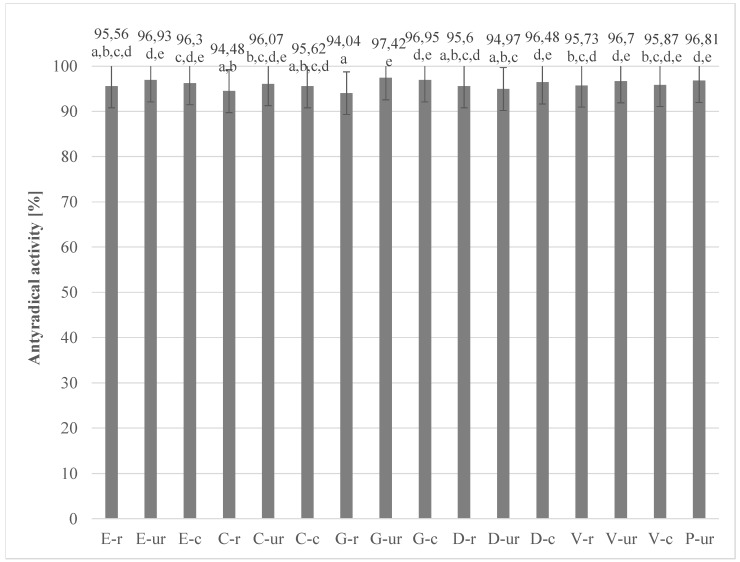
Anti-radical activity of grain extracts from different regions of the world (the same letter means that there are no statistically significant differences between the analysed products at a confidence level of α = 0.05; the abbreviations used in the graph are described in Table 1).

**Figure 5 antioxidants-08-00283-f005:**
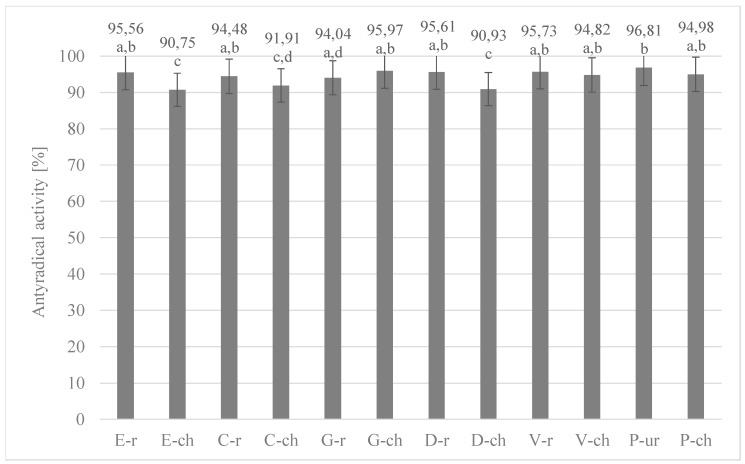
Comparison of anti-radical activity of grain extracts from different regions of the world and of the chocolate extracts produced from them (the same letter means that there are no statistically significant differences between the analysed products at a confidence level of α = 0.05; the abbreviations used in the graph are described in Table 1).

**Table 1 antioxidants-08-00283-t001:** List and labelling of test material No.

No	Abbreviation	Full Name
1	E-r	Ecuador-roasted beans
2	E-ur	Ecuador-unroasted beans
3	E-c	Ecuador-roasted and crushed beanss
4	E-ch	Ecuador-chocolate-roasted beans
5	C-r	Colombia-roasted beans
6	C-ur	Colombia-unroasted beans
7	C-c	Colombia-roasted and crushed beanss
8	C-ch	Colombia-chocolate-roasted beans
9	G-r	Ghana-roasted beans
10	G-ur	Ghana-unroasted beans
11	G-c	Ghana-roasted and crushed beanss
12	G-ch	Ghana-chocolate-roasted beans
13	D-r	Dominican-roasted beans
14	D-ur	Dominican-unroasted beans
15	D-c	Dominican-roasted and crushed beanss
16	D-ch	Dominican-chocolate-roasted beans
17	V-r	Venezuela-roasted beans
18	V-ur	Venezuela-unroasted beans
19	V-c	Venezuela-roasted and crushed beanss
20	V-ch	Venezuela-chocolate-roasted beans
21	P-ur	Peru-unroasted beans
22	P-ch	Peru-chocolate-unroasted beans

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
