# Peer review of "Comparison of the Total Polyphenol Content and Antioxidant Activity of Chocolate Obtained from Roasted and Unroasted Cocoa Beans from Different Regions of the World"

_antioxidants, 2019, doi:10.3390/antiox8080283_

Round 1
Reviewer 1 Report
See attached file

Author Response
Dear Reviewer,
Thank you for your letter enclosing the comments of the reviewers regarding our manuscript entitled: “Comparison of the total polyphenol content and antioxidant activity of chocolate obtained from roasted and unroasted cocoa beans from different regions of the world” (Manuscript ID: antioxidants-551822)
Please see the attachment.
Sincerely

Reviewer 2 Report
Manuscript entitled „Comparison of the total polyphenol content and antioxidant activity of chocolate obtained from roasted and unroasted cocoa beans from different regions of the world" which was submitted to revision in Antioxidants adheres to the journal's standards. The presented data are increasing the knowledge on antioxidant properties of chocolate obtained from raw and roasted cocoa beans from different origins. Thus, the article is suitable for the Special Issue “Antioxidants in Cocoa”. However, manuscript should be improved. Below I put my comments and suggestions:
Abstract
Abstract reflects the content of the article, however the first part of the abstract (page 1 lines 12-16) is rather a short introduction. This sentences should be shortened. In addition, the authors repeat the aim of the study twice (page 1 lines 19-21).
Page 1 lines 22-23, the sentence is unclear and in my opinion, it is not appropriate to use a term "grain" instead of cocoa beans, please revise throughout all text!
Introduction:
In general, the introduction is well-written. However, it seems us to be necessary to make some minor modifications:
page 1 lines 22-23, please correct “cocoa pulp” as “cocoa liquor”.
page 2 line 42, as far as I know monomeric flavan-3-ols (37%), proanthocyanidins (58 %) with various degrees of polymerization, and anthocyanins (4 %) are the major flavonoids found in cocoa beans, please see the articles by Khan et al. 2014, Nutrients. 2014 6(2): 844–880. doi: 10.3390/nu6020844
Please delete the Purpose of the study section and move the goal of the work to the Introduction section.
Materials and Methods:
The experimental design is well described and the analytical procedures were correctly applied. However, there are some points that should be revised prior to publication.
Table 1, the inscriptions in the table should be written in English!
Page 4 lines 107-131, Authors should correct “cm3” as “ml”, please revise throughout all text.
In my opinion, the scavenging effect on the DPPH radical of the samples could be also calculated as the Trolox equivalent's antioxidant capacity or as EC50 value (The EC50 value, defined as the concentration of the sample leading to 50% reduction of the initial DPPH concentration).
Results and discussion:
The authors provide the results obtained with large detail and sometimes the authors repeat the same concepts so the paper it´s heavy to read. There were many good discussion and cited references, however, the writing sentences should be rearranged.
I strongly suggest the authors to try to summarize all the data provided in the manuscript and give a clear idea of the results from this manuscript.
English style should be revised and improved. Some sentences are confusing.
Authors should unify the font throughout the manuscript (including tables and literature)!
Author Response
Dear Reviewer,
Thank you for your letter enclosing the comments of the reviewers regarding our manuscript entitled: “Comparison of the total polyphenol content and antioxidant activity of chocolate obtained from roasted and unroasted cocoa beans from different regions of the world” (Manuscript ID: antioxidants-551822) .
Please see the attachment.
Sincerely
Bogumiła Urbańska
Jolanta Kowalska
